# A Simple and User-Friendly Method for High-Quality Preparation of Pollen Grains for Scanning Electron Microscopy (SEM)

**DOI:** 10.3390/plants13152140

**Published:** 2024-08-01

**Authors:** Aleksey Ermolaev, Majd Mardini, Sergey Buravkov, Natalya Kudryavtseva, Ludmila Khrustaleva

**Affiliations:** 1Center of Molecular Biotechnology, Russian State Agrarian University—Moscow Timiryazev Agricultural Academy, 49 Timiryazevskaya Str., Moscow 127550, Russia; ermol-2012@yandex.ru (A.E.); mr.majdmardini@gmail.com (M.M.); natalia_kudryavtseva@outlook.com (N.K.); 2All-Russian Research Institute of Agricultural Biotechnology, 42 Timiryazevskaya Str., Moscow 127550, Russia; 3Faculty of Fundamental Medicine, M. V. Lomonosov Moscow State University, Moscow 119991, Russia; buravkov@fbm.msu.ru; 4Institute for Biomedical Problems, Russian Academy of Sciences, Moscow 123007, Russia

**Keywords:** SEM, pollen grains, hexamethyldisilazane, shrinkage, folding/unfolding pathways, dicotyledons, monocotyledons

## Abstract

Pollen is becoming an increasingly important subject for molecular researchers in genetic engineering, plant breeding, and environmental monitoring. To broaden the scope of these studies, it is essential to develop accessible methods for scientists who are not specialized in palynology. The article presents a simplified technical procedure for preparing pollen grains for scanning electron microscopy (SEM). The protocol is convenient for any molecular laboratory due to its small set of reagents, ease of execution, low cost, does not require special equipment, and takes only one hour to complete. The high penetrating ability of formaldehyde and the final delicate dehydration using hexamethyldisilazane (HMDS) instead of critical point drying allow for sufficient preservation of the architecture of the aperture, which is considered a gateway for the passage of biomolecules. The method was successfully applied to pollen grains of representatives of dicotyledons (beetroot, petunia, radish, tomato and tobacco) and monocotyledons (lily, onion, corn, rye and wheat). Species studied included insect-pollinated (entomophilous) and wind-pollinated (anemophilous) species. A comparative analysis of the sizes of fresh living pollen grains under a light microscope and those prepared for SEM showed some shrinkage. Quantitative analysis of the degree of pollen grain shrinkage showed that this process depends on the initial shape of dry pollen grains, and the number and structure of apertures. The results support the theoretical model of the folding/unfolding pathways of pollen grains.

## 1. Introduction

Growing interest in pollen as a unique plant material for DNA genotyping [1,2], recombination rate studies [3,4], allergology [5] and as a carrier of macromolecules for gene editing [6,7] has attracted attention to pollen analysis methods. Pollen consists of microscopic grains that contain male gametes capable of fertilizing the female ovule, resulting in the formation of a seed. In plant breeding, pollen morphology is usually evaluated for fertility by light microscopy. However, this technique does not give a perfect image of the pollen grains surface texture. Scanning electron microscopy (SEM) produces detailed magnified images of the pollen grains by scanning their surface. Commonly, SEM analysis of pollen morphology is used in palynology and plant taxonomy as a diagnostic character of a taxon [8,9,10]. The palynological database PalDat (www.paldat.org, accessed on 24 May 2024) of the University of Vienna, Austria, provides free access to SEM images of pollen from a wide range of plant taxa and descriptions of pollen morphology and ultrastructure according to nomenclature and terminology developed by palynologists [11].

The use of SEM as a tool for studying pollen morphology dates back more than half a century, and during this time, several methods for preparing plant material have been proposed. SEM samples are exposed to vacuum conditions and an electron beam which requires an adequate sample preparation (dehydration and coating). All water must be removed from the samples because the water would vaporize in the vacuum. The samples need to be made conductive by covering with a thin layer of conductive material, such as gold.

In earlier studies using SEM, an acetolysis method was used for pollen preparation. The method involves pre-treatment with an acetolysing mixture that comprises acetic anhydride mixed with concentrated sulfuric acid in the ratio 9:1 [12]. Acetolysis dissolves any tissue and removes lipids and debris from the pollen grains. However, such harsh processing may result in a loss of aperture structure [13] or damage and even destruction of pollen grains with thin and/or fragile exines [8]. In addition, in acetolyzed pollen grains, due to the removal of non-resistant structures, there is a significant loss of detailed information at high magnifications [13]. Traditionally, the acetolysis method was used for fossil pollen and did have some marked disadvantages for the study of pollen of living taxa. Later, another technique that avoided the use of acetolyzed pollen was developed [13]. The technique included hydration in Aerosol-OT (sodium bis(2-ethylhexyl) sulfosuccinate), sonication in 1:1 acetone/water, dehydration in ethanol, transfer to amyl acetate, and critical point drying with carbon dioxide. A method of preparing live pollen using dehydration in acidified 2, 2-dimethoxypropane and without previous chemical fixation was proposed by Halbritter [8]. The method is simple, gives good results, and is readily used by palynologists [14]. All the methods mentioned involve a critical point drying of pollen before coating with gold for SEM. The critical point drying method is used to prevent significant irreversible distortion of the sample structure due to the surface tension that occurs when dehydrating liquids (acetone or alcohol) evaporate in the air. The principle of the method is to replace acetone (or alcohol) with liquid carbon dioxide and transfer it to a critical point, at which carbon dioxide instantly turns into a gaseous state. This eliminates the destructive effect of surface tension. Equipment for drying at the critical point is rather expensive. Moreover, the critical point drying of biological samples may cause shrinkage [15,16,17]. Using a protocol of critical point drying, a decrease in the equatorial diameter of pollen in polar view of up to 52% was observed [18].

Hexamethyldisilazane (HMDS) can be a good alternative to replace the critical point drying step in sample preparation for SEM. It was shown on plant and animal cells that final drying by evaporation of HMDS gives the same result as drying at the critical point [16,17,19,20]. HMDS as a final drying agent for the preparation of pollen grains for SEM was proposed by Chissoe and co-authors back in 1994 [19]. Unfortunately, HMDS has been forgotten when preparing pollen for SEM analysis. The advantage of HMDS using is low cost because requires no special equipment, takes only a few minutes and less effort. HMDS is organosilicon monomer freely mixable with alcohol and acetone and polymerizing with no effect on surface tension which is critical during drying samples for SEM and could result in deformation of structure. HMDS also cross-links proteins, therefore adding strength to the sample during air-drying [16].

Biological SEM samples require chemical fixation to preserve and stabilize their structure so that the sample is stable under high vacuum conditions. Glutaraldehyde is commonly used to fix pollen grains, but the fixation may cause some shrinkage [21]. Formaldehyde in concentrations below 5% causes very little initial shrinkage, as was shown on amebocytes [22]. Aldehyde fixatives generally cross-link proteins to maintain the cellular and organelle structures. Also, pollenkitt covering the pollen can masks the fine structure of its surface. Thoroughly removing the pollenkitt could result in a more clear view of the pollen surface morphology. Pollenkitt mostly consists of lipids [23]. It was reported earlier for human tissues that formaldehyde (in formalin) can change the solubility and can lead to the partial disappearing of lipids [24,25,26,27,28]. Thus, formaldehyde can participate in washing out the pollenkitt from the pollen surface.

Mature pollen grains of angiosperm flowers in the open anther are exposed to a dry environment and dehydration. To protect themselves from significant water loss and desiccation, pollen grains have the striking property of folding. Given the importance of this process in the survival of pollen, Roger P. Wodehouse coined the term harmomegathy [29]. Katifori and co-workers attempted to elucidate how pollen wall structure influences harmomegaty [30]. The authors found that aperture morphology plays a critical role in achieving a predictable and reversible folding pattern. Recently Anže Božic and Antonio Šiber [31] with their elastic model of the pollen swelling and bursting added to mechanical theory of Katifori and co-workers. The authors present a broader picture of the interaction of pollen elastic parameters, the number of apertures, their shape and size, which together influence the pathways of pollen folding/unfolding. A broader interpretation of harmomegathy is given in National Agricultural Library Thesaurus Concept Space, U.S. department of Agriculture: “Harmomegathy is the process by which pollen grains and spores change in shape to accommodate variations in the volume of the cytoplasm caused by cells changing hydration”. Thus, pollen is a dynamic mechanical system, the shape and volume of which depend on the environment.

Modern research is based on a multidisciplinary approach aimed at a comprehensive analysis of the relationships between structure and function and understanding of complex life processes. This modern approach brings new challenges to pollen SEM analysis in order to create a richer picture that includes not only qualitative data but also quantitative ones. It is obvious that new or modified existing methods for preparing samples for SEM are needed that would allow obtaining images close to intact shape and size of mature pollen, and at the same time would be simple and user-friendly.

The goal of these studies was to develop a simple, non-labor-intensive protocol that does not require special equipment and at the same time provides high-quality preparations for SEM. In this study, we revealed that formaldehyde resulted in better-quality SEM images when compared to glutaraldehyde as a fixative agent. We used HMDS for final drying pollen grains before coating with gold. The new protocol was tested on various pollen grains of wind-pollinated plants (wheat, rye, maize) and insect-pollinated plants (onion, beetroot, lily, radish, tobacco, petunia, tomato). We paid special attention to the aperture, which is considered as a gateway for the delivery of exogenous biomolecules [6,7]. As was shown in maize, the success of CRISPR/Cas9 delivery into pollen grain directly depends on knowledge of the aperture structure and its physiological state (open or closed) [7]. Aperture is the area of the pollen grain cell wall that has reduced exine deposition, or lack it completely. The aperture patterns are species specific and highly variable. Apertures can differ in shape (e.g., elongated furrows—colpus or sulcus, circular pores with or without lid or a combination of the two—colpori), in number (from zero to many), in size, positions, orientation and ornamentation [32,33,34]. We provided quantitative analysis of the shrinkage rate of pollen grains after SEM sample preparation by comparison with intact pollen grains. Using recent advances in understanding the mechanics of harmomegathy, the results are discussed.

## 2. Materials and Methods

### 2.1. Plant Materials

Pollen of bulb onion (*Allium cepa* L.) cv. ”Myachkovsky 300” was collected on the experimental field of the Molecular Biotechnology Center, Russian State Agrarian University—Moscow Timiryazev Agricultural Academy. Pollen grains of other species was provided by N.N. Timofeev Breeding Station, All-Russian Research Institute of Agricultural Biotechnology and All-Russian Center for Plant Quarantine. Fresh mature pollen grains were used directly in processing for SEM analysis.

### 2.2. Method for Preparing of Pollen Grains for SEM

Reagents and consumables

2.5% formaldehydePrepare freshly before using. 2.5 g of paraformaldehyde is dissolved in 100 mL of 0.1 M PBS using heat (60 °C) and a magnetic mixer (150 RPM). Several drops of NaOH (1 M) are added periodically to facilitate the dissolution. After dissolution pH is adjusted to 7.4.2.5% glutaraldehyde: 5 mL of 50% glutaraldehyde is dissolved in 95 mL of 0.1 M PBS0.1 M PBS (Phosphate-Buffered Saline) buffer8 g NaCl, 0.2 g KCl, 2.2 g ${\mathrm{Na}}_{2}$${\mathrm{HPO}}_{4}$·7$H_{2}$O and 0.26 g $K_{2}$${\mathrm{HPO}}_{4}$ are diluted in 100 mL of ${\mathrm{ddH}}_{2}$O70%, 90% and 96% ethanolHexamethyldisilazane (CAS: 999-97-3; Sigma-Aldrich, St. Louis, MO, USA, Merck, Darmstadt, Germany)CellTrics^TM^ filters (Sysmex, Kobe, Japan)

Pollen preparation

Pollen grains are collected in Petri dishes with a diameter of 5 cm2.5% formaldehyde (or 2.5% glutaraldehyde) is added to a Petri dish with pollen grains and left for 10 min (Appendix A).The pollen grains in 2.5% formaldehyde (or 2.5% glutaraldehyde) are filtered through a filter funnel with an appropriate grid size (Appendix A), so that the pollen grains remain on the filter grid (Appendix A).The pollen grains on the filter grid are thoroughly washed by slowly dripping 2 mL of PBS from a pipette onto the grid of the filter (Appendix A).Then pollen grains on the grid are dehydrated by sequential washing with 70%, 90% and 96% ethanol for 3 min each by dripping of ethanol in the same way as described above (Appendix A).After dehydration, the pollen grains on grid are impregnated with HMDS by slowly adding 500 μL of HMDS over 3 min and air dried until complete evaporation for at least 10 min or left overnight (Appendix A).Pollen grains are sprinkled, gently tapping the filter, onto adhesive double-sided carbon tape placed in a Petri dish (Appendix A), or directly onto a stub with pre-glued double-sided carbon tape (Appendix A).Note: HMDS should not be evaporated directly onto double-sided carbon tape because it tends to act as an adhesive/solvent and frequently deposits a fine residue over the pollen surface.The pollen grains are coated with gold in an argon atmosphere before viewing.

The images were viewed under a scanning electron microscope JEOL JCM-7000 (Tokyo, Japan), or JEOL JSM-6380LA at an accelerating voltage of 15 kV and high vacuum. The microscope’s control program was used to generate digital images.

### 2.3. Measurement of Pollen Grain Size

Pollen grain sizes were measured along the long axis (LA) and short axis (SA) determined on dry pollen grains of ellipsoid shape (Appendix A). In dry pollen of cereals, which has the shape of a truncated pyramid (frustum), in addition to SA at the widest point (${\mathrm{SA}}_{1}$), the narrowest point was measured (${\mathrm{SA}}_{2}$). The complete scheme of LA and SA measurement positions are presented in Appendix A.

Variant 1. Native dry pollen grains were placed on a glass slide and covered with a cover slip without squashing, and then analyzed under a light microscope using oil immersion and objective ×100.

Variant 2. Native dry pollen grains were placed on a drop of immersion oil on a glass slide and covered with a cover slip without squashing, and then analyzed under light microscope using immersion oil and objective ×100.

Variant 3. SEM images.

The measurement of LAs and SAs of pollen grains in Variants 1–2 were performed using ZEN 3.2 program under light microscope Axiolab 5 (Zeiss, Jena, Germany). LAs and SAs of pollen grains on SEM images were measured using DRAWID program [35]. At least 30 pollen grains were measured in each variant.

## 3. Results

### 3.1. Development of a SEM Protocol Using Onion Pollen Grains

We evaluated 2.5% glutaraldehyde and 2.5% formaldehyde fixatives in 0.1 M PBS on onion pollen. SEM showed better result in variant of formaldehyde compared to glutaraldehyde (Figure 1). Formaldehyde was better at removing pollenkitt, a thick layer covering pollen grains that is characteristic of insect-pollinated species. Pollenkitt obscures important details of the pollen surface. In the variant with glutaraldehyde, pollenkitt was not sufficiently removed from the exine surface.

Dehydration was accomplished using washes in graded ethanol solutions (70%, 90% and 96%) for 3 min in each. HMDS was used as a final dehydrating solution. The use of HMDS as a drying agent resulted in a high proportion of essentially distortion-free pollen grains. Our protocol allowed us to clearly visualize the morphology of the pollen grain. Pollen aperture appears as a long furrow (sulcus) extending for half the circumference along the long axis of the pollen grain (Figure 2a). The unfolded aperture was visible in the terminal regions of the sulcus (Figure 2b). The pollenkitt-free surface of the exine ornamentation is finely striated, rugulated and perforated (Figure 2c).

### 3.2. Testing a New Protocol on Pollen Grains of Various Species

We evaluated our method on representatives of insect-pollinated species that have a thick pollenkitt layer and wind-pollinated species with a less pronounced pollenkitt layer.

Application of the developed protocol on insect- and wind-pollinated vegetables and ornamental crops showed good results. 2.5% formaldehyde did an excellent job of cleaning the pollen surface from pollenkitt, allowing the fine structure of the exine to be visualized (Figure 3a–f and Figure 4a,c,e). The aperture architecture was clearly visible without the need for a rehydration step (Figure 3a’–f’ and Figure 4b,d,f).

### 3.3. Effect of SEM Processing on the Shape and Size of Pollen Grains

We wondered how the shape and size of pollen grains changed after treatment according to our SEM protocol. In other words, how close is the SEM image of a pollen grain to the native one in shape and size? We measured long axis (LA) and short axis (SA) on non-squashed preparations of native mature dry pollen grain (1) and maintained in immersion oil (2) under light microscope, using ×100 objective, and on SEM images (3). Comparison of LA and SA measurements of dry pollen grains directly on slides and in immersion oil showed that LA and SA were the same size. Thus, in immersion oil, the shape and size of pollen grains were preserved, while the images were clearer (Appendix A). We summarized the data from variants 1 and 2 to calculate the degree of shrinkage after SEM processing (Appendix A).

The smallest shrinkage (11%), which was the same for LA and SA, was observed in beetroot (*Beta vulgaris*). The original dry pollen grain has a spherical shape (Appendix A), which remains after SEM processing, but in smaller sizes. Beetroot has many apertures evenly distributed over the entire surface of the spherical pollen grain, ensuring symmetrical shrinkage of the pollen grain during dehydration. The plot is clearly illustrated symmetrical shrinkage of the pollen grain after the SEM processing (Figure 5).

The pollen shape of insect-pollinated species with three colpus or colpori apertures and oval shape in the dry state (radish, cultivated tobacco, petunia and tomato) also shifted towards the sphere after SEM processing. The shrinkage of LA varied from 28% to 42%, while SA showed elongation of up to 23% (cultivated tobacco) or remained almost unchanged (radish and tomato).

The pollen shape of Liliaceous species (lily and onion) with elongated oval shape in the original dry state and single sulcus aperture (Appendix A) did not reach the spherical shape in SEM preparation. The transition from an elongated oval shape to an oval shape occurred almost due to the shrinkage of the LA (22% for lily and 28% for onion), while the SA became slightly elongated (Appendix A, Figure 5).

After SEM processing, the frustum-shaped pollen (i.e., truncated pyramide) of wind-pollinated cereal crops changed to oval-shaped (Figure 6). As a result of SEM processing, shrinkage occurred along both axes. Maize had the lowest shrinkage of pollen grains (10% for LA and 12% for SA). Wheat had the highest shrinkage of pollen grains (25% for LA and 31% for SA). It should be noted that pollen grains in all species after SEM processing had irregularly infolded surface.

For all species analyzed, except *Beta vulgaris*, which initially had a spherical shape, a shift towards a more spheroidal shape after SEM processing was observed. The degree of changes LA and SA varied depending on the original shape of the dry pollen grains, architecture of exine and the structure and number of apertures.

## 4. Discussion

### 4.1. Simplified SEM Method

We proposed a simplified method of pollen preparation for SEM. This method provides a clear view of the aperture structure and pollen surface as was shown for pollen grains of representatives of dicotyledons (beetroot, petunia, radish, tomato and tobacco) and monocotyledons (lily, onion, corn, rye and wheat). The species studied included insect-pollinated (entomophilous) and wind-pollinated (anemophilous) species. The protocol is convenient for any molecular laboratory due to its small set of reagents, ease of execution, low cost, does not require special equipment, and takes only one hour to complete. Although our method results in good preservation of surface and aperture structures, pollen shrinkage was not avoided. Shrinkage is mainly the result of dehydration, which is required for a high-resolution SEM under high vacuum.

The evaluation of two fixatives, glutardialdehyde and formaldehyde, showed the benefits of the latter one. The thick layer of pollenkitt covering the surface of onion pollen grains was sufficiently better removed by formaldehyde compared to glutaraldehyde. Formaldehyde small molecule (${\mathrm{CH}}_{2}$O) contains a single aldehyde functional group, whereas glutaraldehyde contains two aldehyde functional groups (${\left({\mathrm{CH}}_{2}\right)}_{3}{\left(\mathrm{CHO}\right)}_{2}$). Monomeric formaldehyde penetrates the tissue quickly. The penetration rate of glutaraldehyde is found to be slower when compared with formaldehyde [36]. Furthermore, formaldehyde can increase solubility of lipids [24,25,26,27,28] and react with some groups in unsaturated lipids [37].

We used the organosilicon monomer hexamethyldisilazane (HMDS) instead of the commonly used ${\mathrm{CO}}_{2}$ drying point for biological samples. This made it possible to obtain high-quality preparations of pollen grains with less time and cost. HMDS has a reduced surface tension and also cross-links proteins, therefore adding strength to the sample during air-drying [16]. The use of HMDS as a final step in the dehydration and drying of pollen samples before SEM was proposed by Chissoe and co-authors in 1994 [19]. Treatment with HMDS increases the charging potential, which is important when using the metal coating method [19].

### 4.2. The Shrinkage Degree after SEM Processing Depends on the Original Shape of the Fresh Dry Pollen Grains, Architecture of Exine, and Structure and Number of Apertures

Pollen grains emerging from the anther undergo rapid desiccation to protect themselves in the environment until reaching the stigma, where they are rehydrated [38]. In fact, pollen grains undergo the processes of rehydration and dehydration during preparation for SEM. At the very beginning, we fix dry pollen grains shaken out of the anther in a 2.5% solution of formaldehyde in 0.1 M PBS. Two processes occur simultaneously, namely, rehydration in an aqueous solution and cross-linking of proteins with formaldehyde. Rehydration is associated with an increase in the volume and unfolding of pollen grains. In the meantime, formaldehyde operates and hurries to preserve the pollen shape and size. Subsequent dehydration at increasing concentrations of ethanol might be one of the major drivers of shrinkage. Recent theoretical studies have highlighted the important role of aperture in folding and unfolding process of pollen grains. The authors showed that folding and unfolding pathways depends on the number and structure of apertures, and the elasticity of the exine. The elastic model was developed by Božič and Šiber [31] for native pollen grains, while we evaluated processed pollen. SEM sample processing is a combination of rehydration (swelling) and subsequent dehydration (shrinkage) overlaid by the processes of fixation with a reduced flexibility of the pollen grain wall. Based on this, we were trying to figure out the acting principles behind pollen shrinkage after the SEM sample preparation in each specific case.

Minimal shrinkage (11%) was observed for beetroot (*Beta vulgaris*). The species has many pores (pantoporate), distributed nearly uniformly on the grain surface. The mechanical model of the pollen grain predicts that the pores deform significantly more than the exine as the grain swells [31]. The authors said that as the number of pores increases, the critical volume of bursting increases and deviates from the values of monoporate grains. This could be explained by an effective elastic interaction between the pores. This effect occurs only for sufficiently large pore opening angles. For instance, the pore angles of pantoporate *Stellaria aquatica* are about $\theta_{0}\approx 0.18$ [34] and monoporate *Zea mays* $\theta_{0}\approx 0.06$ [39]. We can assume that due to such elasticity of the pores and their rapid reaction to rehydration during fixation, the volume of the pollen grain increases. Cross-linking of proteins with formaldehyde fixes the increased volume to some extent. Subsequent dehydration causes shrinkage of the fixed pollen grain, which affects the degree of shrinkage.

The most variable data on shrinkage were obtained for dicotyledonous species with three apertures (colpus or colpori), namely, radish, cultivated tobacco, petunia and tomato. The apertures located at the pollen equator, the most common pattern in this clade, may provide a better compromise between different pollen characteristics [40]. Recent research on colpate pollen showed that different pollen grains sharing the same number and type of apertures can nonetheless fold in quite diverse fashions. Different pathways of folding and unfolding of pollen grains in the analyzed species led to different degrees of shrinkage. But in all species we observed unfolded apertures in SEM images, which indicate rehydration and maintenance of unfolded aperture by cross-linking of proteins with formaldehyde. This fixed unfolded aperture persisted after dehydration [41].

As was shown, folding pathways of pollen grains can be drastically influenced by the shape, size, and number of apertures [41]. Lily and onion, representatives of monocots with a single sulcate aperture extending along the LA, are characterized by a specific folding/unfolding pathway [30]. During rehydration, aperture membrane is primarily exposed to stretching. In this type of equatorially elongated aperture, the entire pollen wall may be involved in stretching and contraction. Thus, in pollen grains subjected to SEM processing, we observe an elongation of SA and shortening of LA, the degree of which depends on the elasticity of the pollen wall. In lilies, pollen grains with a pronounced exine sculpture have less elasticity than onion (22% for lily vs. 28% for onion, see Appendix A).

Wind-pollinated grains showed bizarre exine folding structures after SEM processing. The same irregularly infolded surface was observed on SEM images of pollen grains in the grasses *Elymus arenarius* [42] and *Eleusine indica* [43]. However, for *Eleusine indica* the authors did not use pre-treatment and directly looked at dry pollen coated with gold at a voltage of 10 kV. Ulcerate pollen is a characteristic of most of the species in the Poaceae family (grasses) [44]. The pollen grains with sufficiently hard and small pore (relative to surface area of pollen grain) apertures will not act as a locus of localized deformation, but will instead cause the deformation of the entire grain [41]. It was reported that circular pore apertures perform much worse at guiding pollen folding along a regular pathway compared to elongated sulcus or colpus type apertures [30,41]. The original dry pollen grains had several deep folds along the LA and were shaped like a truncated pyramid (frustum) with an expansion towards the pole, where the aperture is located (Appendix A). After the SEM processing, which involved rehydratation and dehydration, the pollen grains did not restore the original folding shapes.

## 5. Conclusions

The increasing interest in pollen as a supervector leads to the demand for a convenient and reliable method, available for execution in molecular laboratories, for the preparation of samples for SEM. We have adapted a method using HMDS for the preparation of pollen for SEM and developed a new simplified, non-labor-intensive protocol that takes about one hour from collecting pollen grains to observing them under a scanning electron microscope. A short fixation time (10 min) due to the high penetrating ability of formaldehyde and the final delicate dehydration using HMDS allow for sufficient preservation of the architecture of the aperture, which is considered a gateway for the passage of biomolecules. However, it was not possible to avoid shrinkage of pollen grains due to dehydration, which is a prerequisite for conducting analysis under high vacuum conditions that provide good resolution. Quantitative analysis of the degree of pollen grain shrinkage showed that this process depends on the initial shape of dry pollen grains, the exine structure, the number and structure of apertures, which confirms the theoretical model of the folding/unfolding pathways of pollen grains.

## Figures and Tables

**Figure 1 plants-13-02140-f001:**
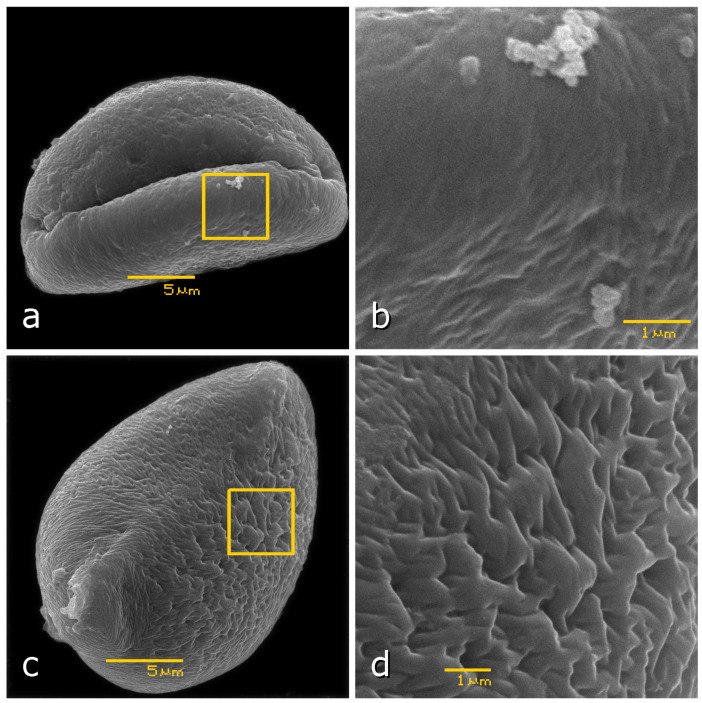
Scanning electron microscopy (SEM) images of *Allium cepa* (bulb onion) pollen grains prepared using glutaraldehyde (**a**,**b**) and formaldehyde (**c**,**d**) protocols. Images (**b**,**d**) represent optical magnification of regions on images (**a**,**c**), respectively, framed in yellow. Scale bars represent 5 μm and 1 μm for images (**a**,**c**) and (**b**,**d**), respectively. The images were taken under a scanning electron microscope JEOL JSM-6380LA. All pollen terms were defined in accordance with the illustrated guide from PalDat (www.paldat.org/static/illustrated_pollen_terminology_2020.pdf, accessed on 24 May 2024).

**Figure 2 plants-13-02140-f002:**
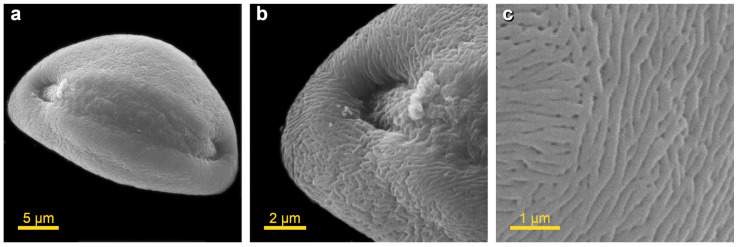
SEM images of *Allium cepa* (bulb onion) pollen grains prepared using developed protocol: (**a**)—polar distal view, aperture (sulcus) extending for half the circumference along the long axis of the pollen grain; (**b**)—unfolded terminal part of aperture; (**c**)—the exine appeared striated, rugulated, perforated. The images were taken under a scanning electron microscope JEOL JCM-7000. All pollen terms were defined in accordance with the illustrated guide from PalDat (www.paldat.org/static/illustrated_pollen_terminology_2020.pdf, accessed on 24 May 2024).

**Figure 3 plants-13-02140-f003:**
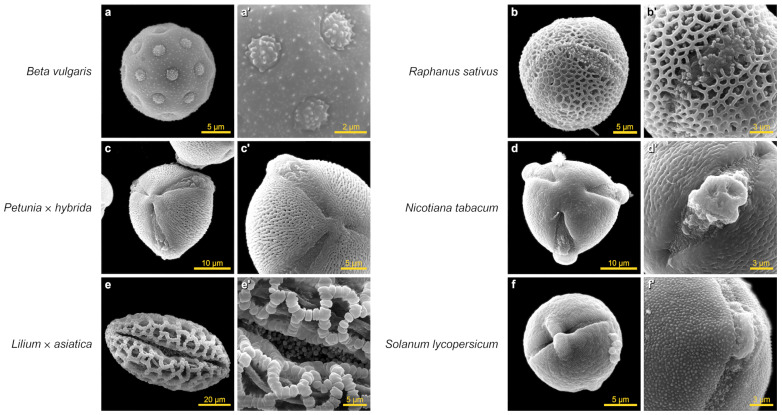
SEM images of pollen grains of insect-pollinated vegetable and ornamental crops obtained using the developed protocol (the images were taken under a scanning electron microscope JEOL JCM-7000). All pollen terms were defined in accordance with the illustrated guide from PalDat (www.paldat.org/static/illustrated_pollen_terminology_2020.pdf, accessed on 24 May 2024). *Beta vulgaris* (beetroot): (**a**)—pollen grain, (**a’**)—aperture type porus, aperture—pore with lid (operculum); exine—echinate; *Raphanus sativus* (radish): (**b**)—pollen grain with three apertures (tricolpate) (**b’**)—aperture type colpus (groove), exine—reticulate; *Petunia* × *hybrida* (petunia): (**c**)—pollen grain with three apertures (tricolporate), (**c’**)—aperture sunken, type colpori, a longitudinally elongated ectoaperture (a colpus) and a transversely elongated endoaperture (an os), exine—rugulate, perforate; *Nicotiana tabacum* (cultivated tobacco): (**d**)—pollen grain with three apertures (tricolpate), (**d’**)—aperture sunken, type colpori (combination pore and colpus), a longitudinally elongated ectoaperture (a colpus) and a transversely elongated endoaperture (an os), exine—rugulate, perforate; *Lilium* × *asiatica* (the Asiatic lily): (**e**)—sulcate pollen grain (monocolpate), (**e’**)—partially unfolded middle aperture area, exine—reticulum cristatum, gemmate, reticulate; *Solanum lycopersicum* (tomato): (**f**)—equatorial view showing an ectexine bridge in the midpoint of the colpus, (**f’**)—view of the aperture after rupture of the equatorial bridge.

**Figure 4 plants-13-02140-f004:**
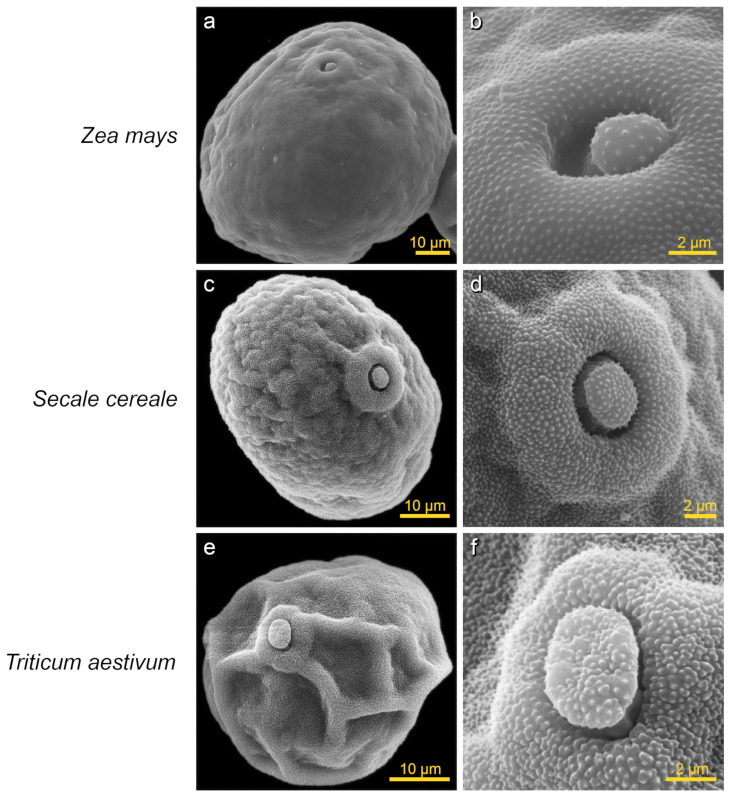
SEM images of pollen grains from cereal crops (Poaceae family) obtained using the developed protocol (the images were taken under a scanning electron microscope JEOL JCM-7000). All pollen terms were defined in accordance with the illustrated guide from PalDat (www.paldat.org/static/illustrated_pollen_terminology_2020.pdf, accessed on 24 May 2024). *Zea mays* (maize, corn): (**a**)—pollen grain with single aperture (ulcerate), (**b**)—aperture type ulcus, peculiarities—operculum, annulus (ring), exine—microechinate; *Secale cereale* (rye): (**c**)—pollen grain with single aperture (ulcerate), (**d**)—aperture type ulcus, peculiarities—operculum, annulus (ring), exine—microechinate; *Triticum aestivum* (bread wheat): (**e**)—pollen grain with single aperture (ulcerate), (**f**)—aperture type ulcus, peculiarities—operculum, annulus (ring), exine—microechinate.

**Figure 5 plants-13-02140-f005:**
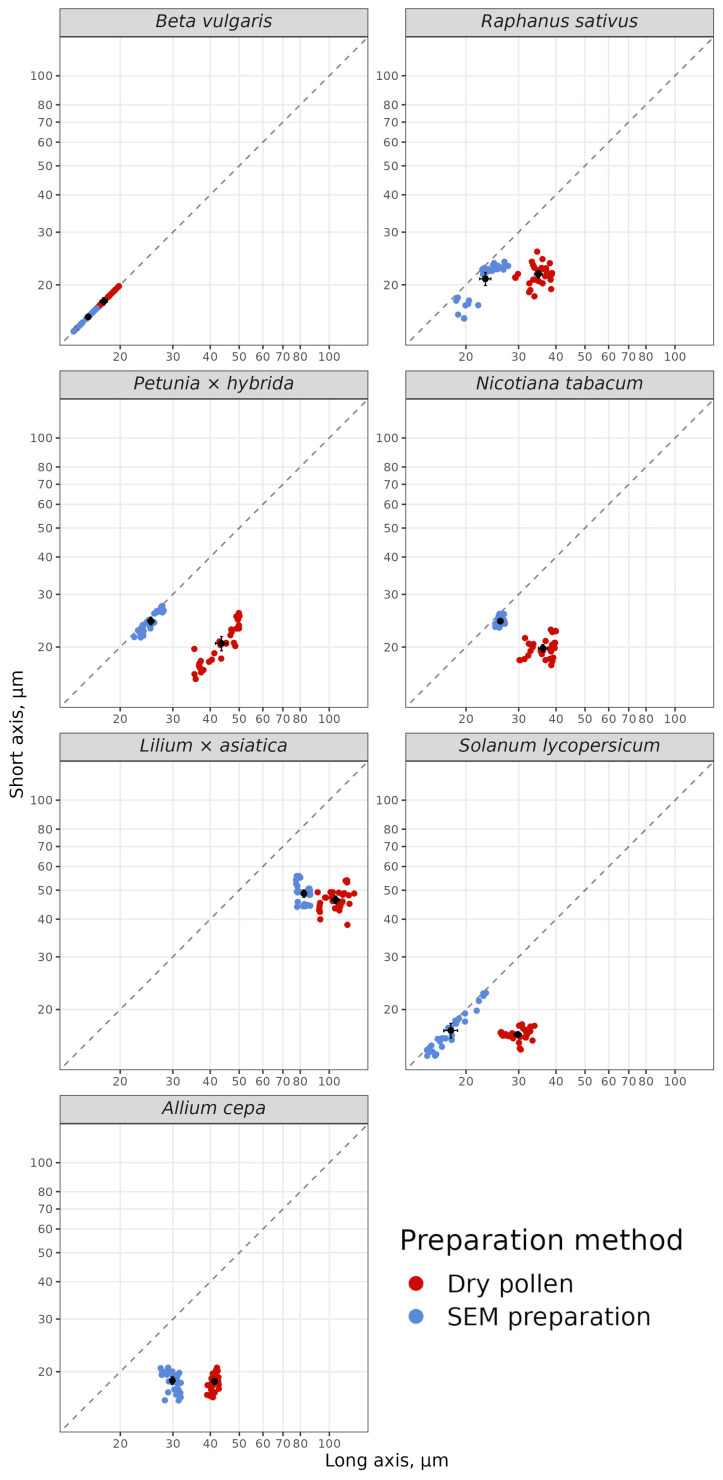
Effect of SEM processing on long axis (LA) and short axis (SA) of pollen grains of insect-pollinated vegetable and ornamental cereals crops. The dotted line on the graph is the ratio SA/LA of the two diameters equal to 1, i.e., pollen grain shape—sphere.

**Figure 6 plants-13-02140-f006:**
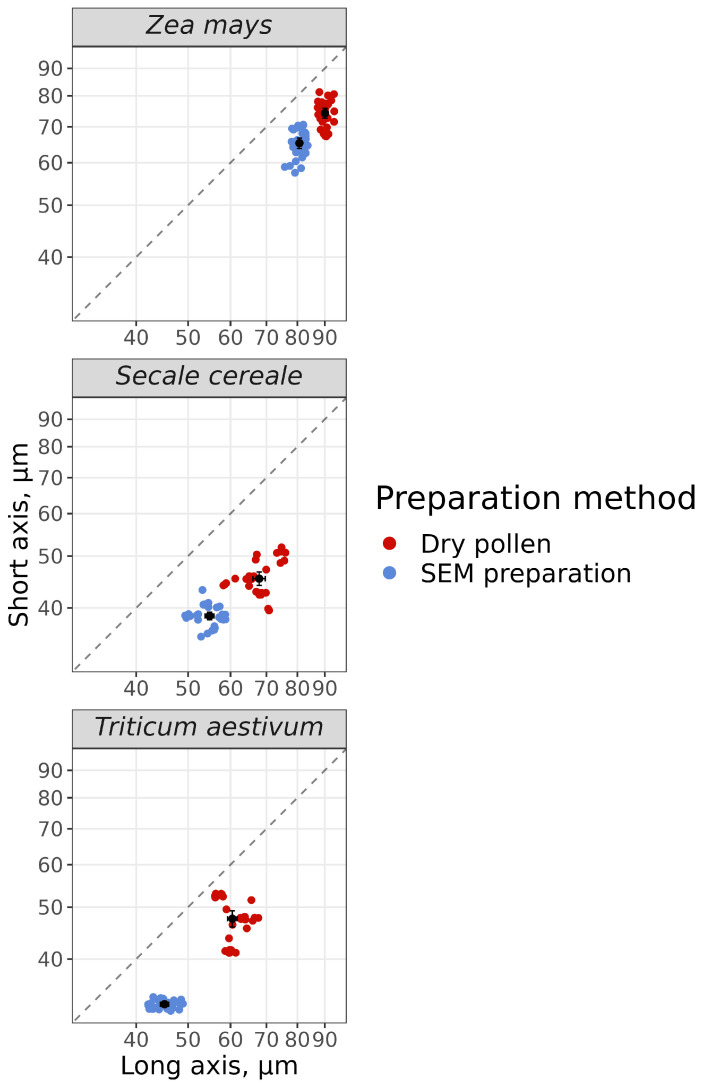
Effect of SEM processing on long axis (LA) and short axis (SA) of pollen grains of wind-pollinated crops. The dotted line on the graph is the ratio SA/LA of the two diameters equal to 1, i.e., pollen grain shape—sphere.

## Data Availability

Data is contained within the article or Appendix A.

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
