# Peer review of "A Simple and User-Friendly Method for High-Quality Preparation of Pollen Grains for Scanning Electron Microscopy (SEM)"

_plants, 2024, doi:10.3390/plants13152140_

Round 1

Reviewer 1 Report

Comments and Suggestions for Authors

A simple and user-friendly method for high-quality preparation of pollen grains for scanning electron microscopy (SEM)

Ermolaev et al.

In the paper, the authors present a simple method for the preparation of pollen grains for scanning electron microscopy. HMDS is used as the drying medium. The method is easy to use and does not require expensive equipment. It can be applied quickly and provides excellent results with regard to the surface structures of pollen grains, proven with pollen grains from insect pollinated plants and from wind pollinated plants.

Though HMDS as a drying agent for SEM preparation is known for several decades the paper gives a new perspective on this topic.

In principle the paper is easy to understand and well written. However, a thorough revision is necessary before the manuscript is published.

The common thread running through the individual passages is sometimes interrupted. At times, individual sentences, often taken directly from the original literature, seem to be randomly strung together - without any immediate context to the text before or after.

The introduction should be revised: pollen grains, scanning electron microscopy, various preparation methods for pollen grains, HMDS, variants in fixation (glutaraldehyde versus formaldehyde), folding pathways (introducing the term harmomegathy).

Material and method are incomplete. The comparative fixation with glutaraldehyde is missing in the description.

Results and discussion have to be clearly separated and corresponding content should be transferred to the introduction or the discussion.

The results support the theory on folding pathways. But the authors should emphasize, that the processed pollen grains are not native pollen grains and that the effect of rehydration and dehydration is overlaid by the process of fixation. 

Further comments on the manuscript

15–17: the exine structure was not part of the investigation; the results “support” the model of folding pathways of pollen grains

26-29: Pollen ultrastructure can be studied using scanning electron microscopy [8-10].

35: not “old” methods; better: existing and proved methods

38-41: SEM samples are exposed to vacuum conditions and an electron beam with the need of adequate sample preparation (drying and coating).

58: All mentioned methods are involved …

61-62: All mentioned references deal with animal or human cells but not plants cells supported by a cell wall. This should be mentioned or references concerning shrinkage with plant material should be cited.

64: “infrastructure”?

68-69: HMDS is organosilicon monomer….. what it the relevance of polymerization to PMDS in this context?

115: 500 mL?

123-124: skip the second part of the sentence (it is clear that SEM uses vacuum and it was already explained in the introduction why dehydration is necessary.  

149-155: not part of results; please transfer to the introduction.

149-160 (including Figure 1): glutaraldehyde is missing in Material and Methods; pollenkitt is washed away during dehydration steps using acetone or alcohol; the fixation solution does not resolve pollenkitt!

Figure 1: other images should be chosen where the differences are clearly to be seen; the main difference between b and d is a different perspective but no differences in details; pollenkitt?

169-182: not part of results; please transfer to the introduction

182: palynologists

184-186: pollenkitt is removed after fixation with formaldehyde and subsequent dehydration in ethanol

186ff: Figure 4?

204: the plot graph is clearly demonstrates

220: “frustum-shaped” is not a common term; please explain it here or use another term

220-224: when talking about shrinkage the minus sign can be skipped

225: Shar peis are dogs that are not known to everyone, please use another comparison

236-237: transfer to discussion

245: the last sentence does not fit here at all.

248-249: what is the relevance of this sentence here?

250-254: two things are mixed; pollenkitt and removal of it (not by the fixative but by the subsequent dehydration using solvents) and penetration of the fixative

256: “more mobile formaldehyde molecules” what does this mean?

265-268: Acetolysis and Tempfix – does not fit into the context of the paragraph

269-272: the authors should start the general discussion with this paragraph

273-317: This part has to be substantially shortened; the processes are folding and swelling; [34, 35]. The authors should be careful with the interpretation. Processed pollen grains are not native. There is a combination of rehydration (swelling) and subsequent dehydration (shrinkage) but overlaid by the processes of fixation with a reduced “mobility” of the pollen grain due to rehydration/dehydration.

340-342: “hysteresis” is a term from systems biology but cannot be applied here because fixation possibly “fix” the system – so it will never be able to go back completely.

348: “super” – use another word

410, 416: Journal of Microscopy

423: Comparative Cytogenetics

433: Microscopy Goday

437: The reference cannot be found in scientific databases; is it correctly cited?

Reviewer 2 Report

Comments and Suggestions for Authors

Review of the manuskript entitled „A simple and user-friendly method for high-quality preparation of pollen grains for scanning electron microscopy (SEM)” 

General concept comments

The results presented in this paper support the main conclusion, which is a simplified technical procedure for preparing pollen grains for scanning eleectron microscopy. The number of investigated species is adequate for the research question discussed, but I am not sure if the authors considered to choose pollen types that are known to change the shape and is infolding in dry condition compared to the hydrated condition. However, it should be noted, that the use of hexamethyldisilazane as adrying agent for pollen for the use in SEM has already been published by Chissoe et al. (1994), and Braet et al. (2003) and from this angle the article is not very novel. This has to be stated more clearly in the manuscript before it can be published. Also, some paragraphs in the text (results and discussion without citing the reference) must be changed or linked to the original research paper as the text reads more or less the same from these two papers (Chissoe et al. 1994; Braet et al. 2003). Moreover, the palynological part and needs to be checked thoroughly as it contains many serious mistakes and needs to be improved!

Specific comments

Abstract comments

line 5-6: “ease of handling, low cost, lack of specialized equipment, minimal expenditure of time” please rewrite the text, as it is the same wording in the abstract of Chissoe et al. (1994)

Introduction comments

I am missing the original method published by Chissoe et al. (1994) in the introduction. The authors refer to this paper and method on the discussion, but it has to be clearly stated also in the introduction, that the method in the current study is not new and that only the protocol for the use of this chemical, based on Chissoe et al. (1994) was adapted in the present study.

The paper by Braet et al. (2003) also demonstrates the usefullness of hexamethyldisilazane in drying cells for SEM, AFM and TEM on hepatic endothelial cells. This will be also an interesting point for your discussion as it shows, that the use of hexamethyldisilazane in drying cells seems to work good for a variety of biological samples, and not only for pollen.

Line 35-36: Please insert the two references in the text where you write about the methods, and I suggest to write “previous methods” instead of “old methods”

Line 33: please change “ultrastructural analysis” (=TEM) to “morphological analysis” (=SEM)

…the study brings challenges to the morphological analysis… (in SEM you study the pollen morphology). Only if you are breaking the wall you can see inside the sturcture, but still the ultrastructure is term used for TEM studies only.

Line 44-45: which studies are using acetolysis before using HMDS?

Maybe it should also be written more clearly which methods are useful to compare with the hydrated shape of pollen. I don´t think that it makes sense to compare the new protocol with pollen studies using acetolyzed pollen grains, as this changes the shape, size and aperture condition completely. It would be better to just compare the new protocol with those SEM studies using DMP+CPD or Ethanol+CPD, which is the standard method used in palynology since many years now.

Line 49-50: “Furthermore, in acetolyzed pollen grains, the number of details at high magnifications was limited” – this is only described in the paper by Rowley 1973 and it is no longer true today. Therefore, it would be good to include here, that this problem is described by Rowley and also the reason for this problem.

Line 64 and 67-68: What do you mean with infrastructure “No infrastructure differences are found between CPD and HMDS drying”? You can only describe the shape, infoldings, size, aperture condition but not the infrastructure? Also, the dehydration process may lead to shrinkage, foldings and/ or infoldings (like the aperture etc.).

line 65-66: Please add the two references (Chissoe et al. 1994 and Braet et al. 2003) in the introduction for the HMDS drying method, as the text is exactly the same in the original paper by Chissoe et al. 1994).

72-73: I would rather write a new protocol instead of a new method, as the method using HMDS is not new.

Material and Methods

In all Figures, except for Fig.1 the latin names of the investigated species are used, this sould be consistent. Please write in Fig.1 SEM pictures of Allium cepa (bulb onion).

I also don´t understand, why the authors did not try the combination of DMP+HMDS, as DMP is also used for easy and fast dehydration as it cleans the pollen walls from pollenkitt? Was this tested in this study? I think that this would also be of great interest for palynologists routinely using DMP for dehydration processes for SEM and TEM.

Line 89: I recommend to change the title to “SEM protocol for preparing pollen”

In the preparation process the pollen material is filtered with a grid and handled in a petri dish for the dehydration process. Wouldn´t it be much easier to dehydrate the pollen grains in an Eppendorf tube and centrifuge before pipetting the liquid and use the next chemical? This might be also considered for most labs having a centrifuge?

Line 130: Just a comment on the term “truncated pyramid (frustum)” – for the shape of Poaceae pollen the term frustum is not recommended, as a palynologist usually would describe the shape of grass pollen in hydrated condition as spheroidal to elliptic and in dry condition as irregular. Please also check pollen descriptions used in palynology such as e.g., on the pollen database PalDat.

Results

In the section 3.1., I am not quite sure why the authors are comparing the SEM protocol and method with the TEM protocol? Is this comparison made to demonstrate the usefulness of the formaldehyde to avoid shinkage, and that it maintains the internal structure? If this was the intention, then this is not clearly written in this section and should be just referred to the chemical properties used?

Line 164: I also don´t understand what the authors mean with “Our protocol allowed us to clearly distinguish the ultrastructure of the pollen grain”?  This is something you can only do with a transmission electron microscope (TEM) and not with SEM?

Line 167-168: please correct and change the pollen terms to: “exine ornamentation is finely striate, regulate and perforate”

Line 178: What is meant with the term aperture operation? Please use known pollen terms to describe the pollen relevant information.

Line 181: pollen morphology and structure are two different things. The SEM pictures in PalDat are displaying the variation in pollen morphology (Shape, size, aperture type, infoldings in dry condition etc.). in this case I recommend to say: “ .. descriptions of pollen morphology and ultrastructure …”.

Line 184: it is cleaning the pollen surface from pollenkitt …

Figure 3: The pollen type in picture a is not Brassica oleracea (https://www.paldat.org/pub/Brassica_oleracea/303972), but rather Amaranthaceae pollen. Please check the correct species name! Brassica is tricolpate, Amaranthaceae are pantoporate. Please correct throughout the manuscript where you used Brassica or cabbage pollen in the text! Please avoid terms like “pimply echinate” it is just “echinate” and you can describe the distribution, size, and length of the echini.

The pollen type in picture b is also not Brassica oleracea! Please check carefully, as the two pollen grains you compare with the different preparation methods cannot be the same type!!!

Figure 3, picture b’: check the picture description, as the picture shows a pollen grin with pores not a colpus

Figure 3, picture c’: the aperture is not sunken in this picture, this is a “normal” colpus

Figure 3, picture 3e: Petunia x hybrida is tricolporate and not tricolpate! (https://www.paldat.org/pub/Petunia_hybrida/305297)

Figure 3, picture 3f’’: it must be: mature pollen grains with three colpori (colporus, plural colpori)

Figure 3, Picture g: please change to: “sulcate pollen grain”.

Figure 4.: cereal crops (family Poaceae)

Figure 4.a-e: the aperture is an ulcus! Please change this in the whole text, as all Poaceae pollen are ulcerate (=aperture on the distal pole), and NOT monocolpate! Please also change the text in the figure legend for all Poaceae aperture descriptions to “aperture type ulcus”

Line 198: The authors state that the use of 45% acetic acid is traditionally used by palynologists? I am working with pollen since more than 15 years (LM, SEM, TEM, recent and fossil) and this is completely new for me. Maybe the authors can give examples when 45% acetic acid is used?

Line 202-237: please change the species Brassica (Cabbage) throughout the text.

Line 207: Broccoli (Brassica oleracea var. italica) is NOT triporate! Correct is tricolpate, as typical for the family Brassicaceae. Please check your plant identification and verify with pollen data published for your investigated plants, as your results are depending on accurate and correct scientific data!

Line 225: Please use pollen terms for descriptions and avoid comparisons of pollen with a dog species. As not everyone knows a Shar Pei.

Line 230-231: Please change to “As a result of rehydration, the shape of the investigated grass pollen was spheroidal and swelling increased their volume”.

For the results I recommend to either ask a palynologist to contribute to the manuscript to achieve better results and pollen descriptions or to reduce the pollen descriptions and just compare the methods by illustrating the pictures with the correct species and pollen types.

Discussion

Line 242-244: please rewrite the text or use the correct reference, as the text is the same in the original paper!

Line 269-272: Please correct the species investigated. You may also write that you used pollen grains that are infolded in dry condition and differ in their shape etc.?

Lines 289-308: Correct the species name and change Brassica oleracea

Line 310: What is meant with “the apertures are collocated at the pollen equator”? I think this should be “The apertures are located at the pollen equator”?

Line 333-334: I do not understand what the authors mean with “sufficiently hard and small pore apertures”?

Line 345-346: This information is superfluous in this context.

Conclusions

Line 350: I would really recommend writing: “We have adapted a method using HMDS for the preparation of pollen for SEM and developed a new preparation protocol” as the method using HMDS for dehydration was not developed by you.

Round 2

Reviewer 1 Report

Comments and Suggestions for Authors

The authors have responded to all comments from the first review and have adapted the manuscript. The manuscript has clearly improved and represents a very good contribution to the topic covered. 

One single point should be slightly changed: 

231 - 244: This first paragraph does not fit with the results and should be incorporated into the introduction - where it fits. 

Author Response

Dear Reviewer,

Thank you again for your careful analysis.

Comments: One single point should be slightly changed: 

231 - 244: This first paragraph does not fit with the results and should be incorporated into the introduction - where it fits. 

Response: Thank you for pointing this out. We have replaced the text to the Introduction according to your recommendations, lines 30-33; 116-124.

Reviewer 2 Report

Comments and Suggestions for Authors

Dear authors, 

the manuscript has become much better now. I congratulate you on the revised version and honor the efforts you put into the revision. 

All the best, 

Author Response

Dear reviewer,

Thank you very much for your contribution to improving our manuscript.